# Ablative or Surgical Treatment for Small Renal Masses (T1a): A Single-Center Comparison of Perioperative Morbidity and Complications

Jari Radros [1,2], Anders Kjellman [1,2], Lars Henningsohn [1,2], Yngve Forslin [1,3], Martin Delle [1,3], Marianna Hrebenyuk [1,2], Anna Thor [1,2], Tomas Thiel [1,2], Maria Hermann [1,2] and Per-Olof Lundgren [1,2,*]

1   Karolinska Institute C1:77, Karolinska University Hospital, 141 86 Stockholm, Sweden; jari.radros@ki.se (J.R.); anders.kjellman@ki.se (A.K.); yngve.forslin@ki.se (Y.F.); martin.delle@ki.se (M.D.); marianna.hrebenyuk@ki.se (M.H.); anna.thor@ki.se (A.T.)
2   Department of Urology, Karolinska University Hospital, 141 86 Stockholm, Sweden
3   Department of Radiology, Karolinska University Hospital, 141 86 Stockholm, Sweden
*   Correspondence: per-olof.lundgren@regionstockholm.se; Tel.: +46-8-12-387-756

**Abstract:** The purpose of this study is to evaluate the treatment safety of thermal ablation compared to surgical treatment of T1a tumors (small renal masses) at a high-volume center. We conducted an observational single-center study based on data collected form the National Swedish Kidney Cancer Register (NSKCR) between 2015 and 2021. In total, 444 treatments of T1a tumors were included. Patients underwent surgery (partial or total nephrectomy) or ablative treatment—radiofrequency ablation (RFA) or microwave ablation (MWA). Patient characteristics were retrieved from patient records, and tumor complexity was estimated from pre-interventional CT scans. The odds ratio (OR) of suffering from a severe surgical complication following ablative treatment was estimated using a logistic regression model adjusted for age, BMI, ASA physical status classification, smoking status and RENAL nephrometry score. The frequency of severe surgical complications was 6.3% (16/256 treatments) after surgical intervention and 2.1% (4/188 treatments) following ablative treatment. Our primary hypothesis that ablative treatment is associated with a lower risk of severe surgical complications is supported by the results (OR 0.39; 0.19–0.79; *p* = 0.013). When adjusting for age, smoking status, ASA score, BMI score and RENAL nephrometry score, we see an even greater difference between the two groups (OR 0.34; 0.17–0.68; *p* = 0.002). Our study was limited by the differences in patient and tumor characteristics between the two compared groups and the study design. If oncological outcomes are found to be comparable, ablative treatment should be considered as a first-line treatment for all small renal masses.

**Keywords:** ablation; surgery; treatment; kidney cancer; intervention

## 1. Introduction

Renal cell carcinoma (RCC) represents the sixth most diagnosed cancer in the world [1,2]. Due to the increased use of non-invasive imaging modalities in higher-income settings and an overall aging population, an increase in the incidence of RCC has been seen over time [3]. In Sweden, 1300 new cases of RCC were reported to the National Swedish Kidney Cancer Register (NSKCR) in 2021, and 41% of these tumors were classified as cT1a. Approximately 49% of patients diagnosed with RCC are 70 years or older (NSKCR).

As many of these patients have significant comorbidities, the need for alternative treatment options to surgery has emerged. Although the recommended treatment for RCC is surgical removal of the tumor (partial or total nephrectomy), the use of minimally invasive methods such as cryotherapy, radiofrequency ablation (RFA) and microwave ablation (MWA) have significantly increased [3].

In 2011, approximately 9% of all T1a tumors in Sweden were treated with ablative therapy compared to 25% in 2021 (NSKCR).

Thermal ablation (TA) treatments have shown promising results in the treatment of small renal masses (i.e., tumors confined to the kidney with size <= 4 cm). Although nephron sparing surgery remains the gold standard procedure for small renal masses, a slightly higher degree of renal impairment has been shown even after partial nephrectomies compared to RFA [4]. In addition to better post-operative renal function, previous reports suggest that ablative therapies may be associated with similar rates of local recurrence-free survival, cancer-specific survival, metastasis-free survival, disease-free survival as well as a significantly reduced occurrence of post-operative complications compared to partial nephrectomies of T1a tumors. However, the evidence is inconclusive [5–7].

According to EAU guidelines, ablative treatment is only recommended for frail patients who are unfit for major surgery in the current situation due to the lack of solid evidence regarding the clinical and oncological effectiveness of TA compared to PN [8].

The aim of this study is to compare the frequency of severe surgical complications between TA and surgical treatment for small renal masses, as well as the duration of hospital stay and unplanned readmissions. Our primary hypothesis is that ablative treatment is associated with a lower risk of severe complications, shorter period of hospital care and fewer cases of unplanned readmissions.

## 2. Materials and Methods

The cohort was defined from an institutional database at a tertiary referral center for renal cell carcinoma where approximately 120–160 new cases of renal cancer are treated every year. This is an observational mono-institutional study based on data collected from the NSKCR, patient records and preoperative/pre-ablation CT scans.

The NSKCR was established in 2005 to evaluate and improve the quality of care for patients with RCC in Sweden. It includes data on diagnosis, tumor characteristics and treatments. The NSKCR is linked to the National Swedish Cancer Registry. The Swedish National Cancer Registry is maintained by the Swedish National Board for Health and Welfare. Inclusion in the NSKCR is voluntary, whereas reporting to the National Cancer Registry is mandatory and hence, coverage is nearly 100%.

In the cohort we identified, all cT1a tumors were treated between 2015 and 2021 at a tertiary referral center. A total number of 460 treatments were identified in our database, either surgery (total or partial nephrectomy) or TA. Patients treated for bilateral renal tumors (seven procedures) as well as patients with tumors in transplanted kidneys were excluded (three procedures). In one case, the patient underwent endovascular intervention due to bleeding complications following a renal biopsy and was not given any further treatments and was therefore excluded from the study. Consequently 444 treatments were included in our study: 188 ablative treatments, 207 partial nephrectomies and 49 total nephrectomies. All treatments were primary interventions, and all patients were naïve to chemotherapy, immunotherapy and radiation therapy. A flow chart of the inclusion/exclusion process is presented in Figure 1. All complications were registered in the NSKCR at least 90 days after surgery. Data was validated in patient records, and tumor complexity was estimated using the RENAL nephrometry score ranging from 4 to 12 points, assessing the tumor sizes, endophytic/exophytic properties, nearness to the collecting system (i.e., the renal pelvis) and the involvement of the renal polar lines.

This study was approved by the regional Ethical Review Board (2020-03839).

The cohort included a total of 435 patients and 453 tumors. Multiple unilateral renal tumors were identified preoperatively and treated in seven patients (five patients had two tumors, and two patients had three tumors).

The manuscript was drafted using the Strengthening the Reporting of Observational Studies in Epidemiology (STROBE) checklist [9], and the guidelines for the reporting of statistics for clinical research in urology [10].

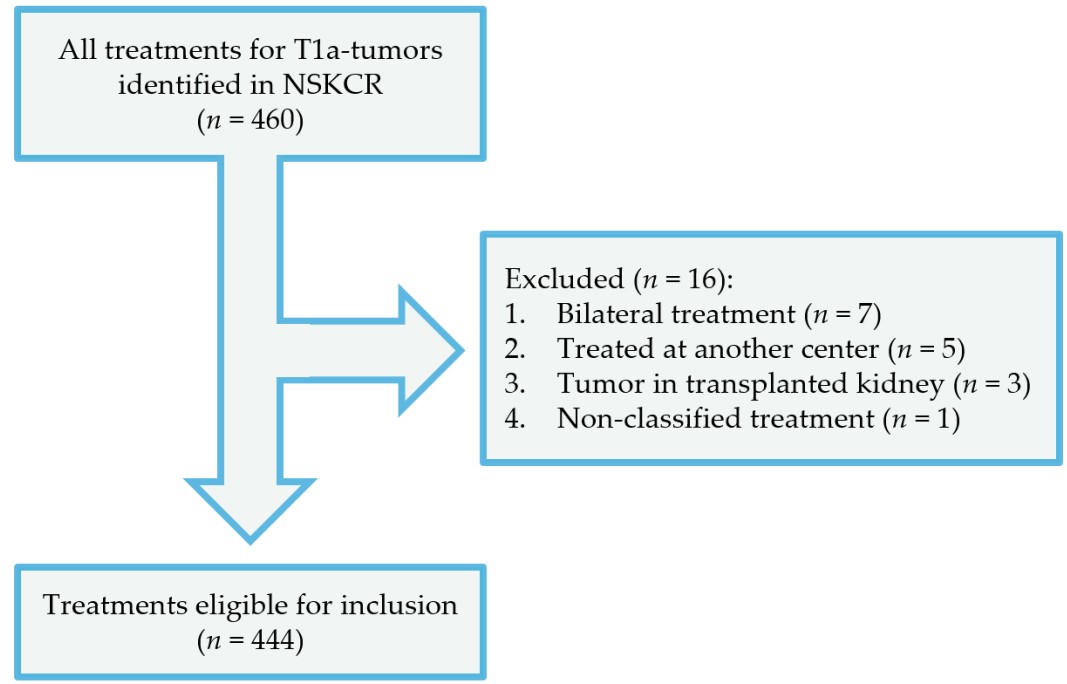

**Figure 1.** Flowchart of the inclusion process.

*2.1. Treatment Modalities*

Partial and total nephrectomies were performed via open surgery, conventional laparoscopy or robot-assisted laparoscopy, as described in Table 1. Surgical treatment was performed with the patient under general anesthesia by consultant urologic surgeons. Open nephrectomy was mainly performed via a subcostal incision. A transperitoneal approach was used in conventional and robot-assisted laparoscopies. In both laparoscopic modalities, gas insufflation of the abdominal cavity was established using an open Hasson technique. Partial nephrectomy hilar control, if deemed necessary, was managed by clamping the main renal artery using one or two bulldog clamps, which were removed after the resection. The kidney/resected tumor was extracted via a sealed specimen bag (Endocatch) through the incision of the umbilical port after the enlargement of the incision to adequate size.

**Table 1.** Surgical treatment modalities.

|  | Open Technique (%) | Conventional Laparoscopy (%) | Robot-Assisted Laparoscopy (%) |
|---|---|---|---|
| Partial nephrectomy (*n* = 207) | 57 (*n* = 118) | 26 (*n* = 53) | 17 (*n* = 36) |
| Total nephrectomy (*n* = 49) | 47 (*n* = 23) | 33 (*n* = 16) | 20 (*n* = 10) |

Patients scheduled for ablative treatment were admitted to the hospital either in the evening the day before treatment or on the day of treatment. TA treatment was performed either with RFA or MWA with the patient under general anesthesia. RFA was the standard TA modality between 2015 and 2018. MWA was introduced in 2019 and from 2020, most TA procedures were conducted using this technique. The tumors were localized via ultrasound or CT scan (standard modality from approximately April 2020), and the ablation needle was placed centrally in the tumor using the respective image guidance. RFAs were performed with the placement of a 17-gauge antenna (Cool-tip™ RF Ablation System E Series, Covidien/Medtronic, Kista, Sweden). In cases treated using RF, the duration of ablation was 12 min, and a temperature of 65 °C directly after ablation was considered adequate. In MWA, a 13-gauge microwave antenna (Emprint Ablation System, Medtronic Kista, Sweden) was placed, and 45–100 W was administered for approximately 5–10 min depending on the tumor size and energy setting. Track ablation was always performed after tumor ablation to minimize the risk of local seeding of tumor cells.

## 2.2. Data Collection

Tumor size was determined using radiologic measurements from the preoperative CT scan. If multiple unilateral tumors were present, the largest tumor was included. Most of our patients received surgical treatment, as seen in Table 2. The group receiving ablative treatment was older (73 vs. 64; $p < 0.001$) and had higher ASA scores (2.6 vs. 2.0; $p < 0.001$) and higher ECOG scores (1.3 vs. 0.6; $p < 0.001$) on average. In the ablation group, tumors were smaller on average (21 mm vs. 26 mm; $p < 0.001$). The surgical treatment group had marginally more complex tumors according to their RENAL nephrometry scores (6.7 vs. 6.2; $p = 0.0013$). No statistically significant difference was seen between the two groups regarding BMI score or smoking status.

**Table 2.** Patient and tumor characteristics.

| | Ablative Treatment ($n = 188$) | Surgical Treatment ($n = 256$) | $X^2$ |
|---|---|---|---|
| Active smoker (%) | 14 | 12 | 0.6 |
| Age (years) (range; mean) | 65–78; 73 | 54–73; 64 | <0.001 |
| Tumor size (mm) | 21 | 26 | <0.001 |
| BMI (mean) | 28 | 27 | 0.5 |
| ASA score (mean) | 2.6 | 2.0 | <0.001 |
| ECOG score (mean) | 1.3 | 0.6 | <0.001 |
| RENAL nephrometry score (%) | | | |
| Low complexity (RS 4–6) | 60.1 ($n = 113$) | 50.0 ($n = 128$) | 0.15 |
| Moderate complexity (RS 7–9) | 38.3 ($n = 72$) | 44.1 ($n = 113$) | 0.4 |
| High complexity (RS 10–12) | 1.6 ($n = 3$) | 5.9 ($n = 15$) | 0.8 |

Surgical complications were defined using the Clavien–Dindo classification with severe surgical complications defined as those equal to or greater than three—i.e., complications requiring surgical, endoscopic or radiological intervention.

## 2.3. Statistical Analysis

Differences in baseline characteristics were estimated using a one-way ANOVA test. Estimations of differences in the outcome variable were estimated via a logistic regression model adjusted for age, BMI, ASA physical status classification, smoking status and RENAL nephrometry score. A $p$-value of $< 0.05$ was regarded as statistically significant.

## 3. Results

The incidence of severe surgical complications (Clavien–Dindo $\geq 3$) was presented and stratified based on treatment method as seen in Table 3. The frequency of surgical complications was 6.3% (16/256 treatments) after surgical intervention and only occurred after partial nephrectomies (7.7%; 16/207 treatments). In the group receiving ablative treatment, surgical complications occurred in 2.1% (4/188 treatments). The most common surgical complications were bleeding and urinary leakage following surgery. Bleeding occurred in 7/256 (2.7%) surgical treatments. Bleeding control was established via open surgery in three cases, transurethral surgery in one case and endovascular intervention in two cases, while in one case, the patient required intensive care and the bleeding ceased spontaneously. In the ablative treatment group, bleeding occurred in 1/188 (0.5%) case, and the patient was treated using an endovascular intervention. In the group undergoing total nephrectomies, we had no cases of severe surgical complications.

The length of hospital stay was four times longer (4.8 days vs. 1.2 days) following surgery compared to ablative treatment, and the rate of readmission was slightly higher in the group receiving surgical intervention compared to ablative treatment (8.2% vs. 7.4%).

The length of hospital stay was 4.3 days in the group undergoing total nephrectomies, and the rate of readmission was 6.1%. The primary causes for unplanned readmissions (within 90 days) are listed in Table 4.

**Table 3.** Length of hospital stay (LOS), rate of readmission and surgical complications. * Requiring urinary catheter.

| | Open Technique ($n$ = 141) | Conventional Laparoscopy ($n$ = 70) | Percutaneous Access ($n$ = 187) | Robot-Assisted Laparoscopy ($n$ = 46) | $X^2$ |
|---|---|---|---|---|---|
| LOS (days) (range; mean) | 3–15; 6.0 | 1–10; 3.8 | 1–9; 1.2 | 1–15; 2.8 | $p < 0.001$ |
| Unplanned readmission (%) | 12.8 ($n$ = 18) | 4.3 ($n$ = 3) | 7.5 ($n$ = 14) | 0 ($n$ = 0) | $p = 0.019$ |
| Surgical complications (Clavien–Dindo $\geq$ 3) (%) | 7.1 ($n$ = 10) | 4.3 ($n$ = 3) | 2.1 ($n$ = 4) | 6.5 ($n$ = 3) | $p = 0.022$ |
| Urinary leakage (%) | 4.3 ($n$ = 6) | | | 2.2 ($n$ = 1) | |
| Bleeding (%) | 2.8 ($n$ = 4) | 1.4 ($n$ = 1) | 0.5 ($n$ = 1) | 4.4 ($n$ = 2) | |
| Urinary retention * (%) | | | 0.5 ($n$ = 1) | | |
| Other (%) | | 1.4 ($n$ = 1) | 1.0 ($n$ = 2) | | |
| Abscess (%) | | 1.4 ($n$ = 1) | | | |

**Table 4.** Causes for unplanned readmissions (within 90 days).

| | Ablative Treatment ($n$ = 118) | Partial Nephrectomy ($n$ = 207) | Total Nephrectomy ($n$ = 49) |
|---|---|---|---|
| Anemia | 1 | | 1 |
| Chest pain | 1 | | |
| Herniated disk | 1 | | |
| Dyspnea | 1 | 2 | |
| Esophagitis | 1 | | |
| Minor trauma | 1 | | |
| Intestinal obstruction | 1 | | |
| Infection | 4 | 6 | 1 |
| Pain | 1 | | |
| Syncope | 1 | | |
| Bleeding | | 1 | |
| Hematuria | | 1 | |
| Melena | | 1 | |
| Abdominal pain | | 2 | |
| Jaundice | | 1 | |
| Kidney stone | | 1 | |
| Fatigue | | 1 | |
| Urinary leakage | | 1 | |
| Vertigo | | 1 | |
| Cholecystitis | | | 1 |
| Other | 1 | | |
| Total | 14 | 18 | 3 |

The logistic regression analysis of severe complications following ablative treatment compared to surgical treatment are presented in Table 5. Ablative treatment was associated with a lower risk of severe post-interventional complications (OR 0.39; 0.19–0.79; $p = 0.013$).

When adjusting for age, smoking status, ASA score, BMI score and RENAL nephrometry score, we see an even greater difference between the two groups (OR 0.34; 0.17–0.68; *p* = 0.002).

**Table 5.** Odds ratio (OR) for severe surgical complications (Clavien–Dindo ≥ 3) following ablative treatment compared to surgical intervention. * Adjusted for age, smoking status, ASA score, BMI score and RENAL nephrometry-score.

|  | OR | 95% CI | *p* |
|---|---|---|---|
| Unadjusted | 0.39 | 0.19–0.79 | 0.013 |
| Adjusted * | 0.34 | 0.17–0.68 | 0.002 |

## 4. Discussion

Regarding severe surgical complications, our results suggest that ablative treatment of small renal masses is safer than surgical removal because of the lower frequency of severe complications. In accordance with current treatment guidelines, this treatment-characteristic is especially required for the elderly and/or frail population with a higher presence of comorbidities [8]. Small renal masses rarely metastasize [7] and hence do not affect cancer-specific survival in an epidemiological context. We will therefore have to base our clinical decision-making, at least in part, on other variables of which the risk for severe complications is one.

Furthermore, the length of hospital stay (LOS) after surgery differs by a factor of four in this material with significantly shorter LOS for patients treated with TA. The magnitude of the difference in LOS is, in part, due to the large number of procedures performed using the open transperitoneal technique. Over time, a higher proportion in the surgical cohort is operated upon using laparoscopic and robot-assisted techniques with a shorter LOS compared to open procedures and thus, as expected, reducing but not eliminating the difference vs. TA (mean LOS for robot-assisted procedures is 2.8 vs. 1.2 days for TA).

Studies suggest that ablative treatments are associated with a higher risk of re-intervention than surgical removal of tumors, but since cancer-specific mortality is low in patients treated for small renal masses, the body of evidence regarding hard endpoints remains small [11]. In this evaluation we have abstained from evaluating differences in oncological outcomes, in part due to the low rates of events in the population at large and in part due to the inherent differences in baseline characteristics in the two treatment arms. Nevertheless, we consider the lack of solid oncologic endpoints a limitation. There are also different thermal ablation techniques with some differences regarding advantages and risks, although the general risk for complications among different ablative techniques has not been shown to differ significantly from each other [12–14]. The heterogenicity in the ablation study group (consisting of both RFA and MWA) could be considered a limitation, but it also makes the comparison with surgery more generalizable. Moreover, with the higher potency of MWA, the transition towards the use of CT guidance could enable better visualization of the tumor as well as facilitation of the proper avoidance of adjacent structures such as the colon, small bowel, ureter and vessels. Our estimation of tumor complexity, the RENAL score, is foremost developed to predict complexity in partial nephrectomies with the surgical approach, and recently, Musi et al. proposed a modified complexity score, the SuNS score, which seems to better predict treatment success in the ablative setting [15]. In our material, however, the more commonly used RENAL score is used to adjust for confounders.

None of the patients in the ablation group were treated with cryoablation, which is considered less damaging to the renal collective system and is therefore advantageous for endophytic or centrally located tumors [12], and this possibly affected the generalizability of the results. Ablative techniques can also be combined with pyeloperfusion to further protect the collective system in centrally localized renal tumors [16].

In conclusion, ablative treatments are associated with a lower risk for severe complications compared to surgery for small renal masses, and if oncological non-inferiority for ablative treatments could be demonstrated, the lower frequency of complications should prompt this alternative to be considered for all patients with small renal masses.

**Author Contributions:** Conceptualization, J.R., A.K., M.H. (Marianna Hrebenyuk) and P.-O.L.; methodology, J.R., A.K., T.T. and P.-O.L.; software, J.R. and P.-O.L.; validation, M.H. (Maria Hermann), L.H., M.H. (Marianna Hrebenyuk), A.T., T.T., Y.F. and M.D.; formal analysis, J.R. and P.-O.L.; investigation, J.R., M.D. and Y.F.; resources, P.-O.L., L.H. and A.K.; data curation, J.R. and Y.F.; writing—original draft preparation, J.R. and P.-O.L.; writing—review and editing, J.R., A.K., L.H., Y.F., M.D., A.T., M.H. (Marianna Hrebenyuk), M.H. (Maria Hermann), T.T. and P.-O.L.; visualization, J.R.; supervision, A.K. and P.-O.L.; project administration, J.R. and P.-O.L.; funding acquisition, P.-O.L. and A.K. All authors have read and agreed to the published version of the manuscript.

**Funding:** This research was in part funded by Grafströms foundation for urological research, Stockholm, Sweden 2023-06.

**Institutional Review Board Statement:** The study was conducted in accordance with the Declaration of Helsinki and approved by the Swedish National Ethics Committee 2020-03839.

**Informed Consent Statement:** Patient consent was waived due to the retrospective nature of the data collected in accordance with the decision from the ethics committee.

**Data Availability Statement:** The data presented in this study are available on request from the corresponding author.

**Conflicts of Interest:** The authors declare no conflicts of interest.

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
