# Peer review of "Ablative or Surgical Treatment for Small Renal Masses (T1a): A Single-Center Comparison of Perioperative Morbidity and Complications"

_curroncol, doi:10.3390/curroncol31020069_

Round 1

Reviewer 1 Report

Comments and Suggestions for Authors

The authors wanted to report the safety of kidney masses thermal ablation.

The topic is of great interest and several reports are now coming in the literature with more reliable series and larger cohorts. Moreover, it is difficult to obtain optimal results without any oncological evaluation, but i do understand the goal of the paper itself.

An Italian group recently published on the ideal size cutoff for these patients. This paper should be cited

More over, my concern is relative to the high number of patients treated with open surgery (57%). This cohort indeed can develop a higher rate of post-op complications compared to lap or robotic. Probably, a sub-analysis should be run in the manuscript to show if the type of surgery can impact on the outcomes.

Lastly, the authors chose the RENAL nephrometry score to report the risk of complications. As we can understand, the approach fot TARF is posterior, and the available nephrometric scores can not exactly predict complication because they are built on the open approach. For this reason the SUNS score has been proposed. Probaly a different analysis should be set to compare the different rate of complications.

Predicting Peri-Operative Outcomes in Patients Treated with Percutaneous Thermal Ablation for Small Renal Masses: The SuNS Nephrometry Score. Musi G, Luzzago S, Mauri G, Mistretta FA, Varano GM, Vaccaro C, Guzzo S, Maiettini D, Di Trapani E, Della Vigna P, Bianchi R, Bonomo G, Ferro M, Tian Z, Karakiewicz PI, de Cobelli O, Orsi F, Piccinelli ML.Diagnostics (Basel). 2023 Sep 15;13(18):2955. doi: 10.3390/diagnostics13182955.PMID: 37761322 

Reviewer 2 Report

Comments and Suggestions for Authors

The authors should be complimented with their work. Their aim is to demonstrate that ablation can be a valid alternative and maybe in future a fist option of treatment in small renal masses. We must of course remember that in surgery, very often the histological report is of great use for characterizing the lesion and future possibile treatments. This advantage is lost in ablation. 

The obtained results show that this may be the case and that if one day oncological outocomes were to be proven as comparable to those of surgery, ablation in its various forms would have to be considered greatly, considering also its low rate of severe complications and its minimal invasivity.

The title is adeguate to the contento of the manuscript. 

The abstract is well structured and contains the necessary information, underlining outcome measures. The introduction is brief and well explained. Used method is also quite clear although not containing full statistical methods that were applied. 

The obtained results are described and well discussed, underlining their strength and possible future applicability.

In future, in order to overcome a few limitations, this study should be extended to other centers, higher number of patients and with a variability of operators.

I believe this study is well written and in line with this journal. It seems to be scientifically acceptable, well written and therefore suitable for publication in present form.

Reviewer 3 Report

Comments and Suggestions for Authors

The aim of the study compare complication rates between TA and surgical treatment fo small renal masses. Topic is innovative, being ablative treatments increased in the las years. However, there are some critical aspects that AIuthors should report in order to improve the overall quality of manuscript. 

- Baseline features should be improved. Charlson Comorbidity Index should be included. Moreover, RENAL nephrometry score should be considered a categorical variable and RENAL risck score should be preffered. 

- Table 2 showed baseline statistically significant differences between cohort. Therefore a propensity scored match analysis must be performed in order to obtain two homogeneous cohort where it souds reasonable to compare postoperative complications rates. 

- Baseline features should be reporting partial nephrectomy and radical nephrectomt (not total) separetely. 

- Pvalue shopul be recorded in table 3

- Finally, univariable and multivariable regression analysis must be performed to identify independet predictors of CD>3 complications, and addressing if approach (ablative vs surgical) could be included. 

Reviewer 4 Report

Comments and Suggestions for Authors

The article by Jari Radros et al. evaluate treatment safety of thermal ablation (radiofrequency ablation and microwave ablation) compared to surgical treatment (partial nephrectomy and Total nephrectomy) of small renal masses, using data from NSKCR.

This article is well organized, nicely written. The reviewer highly appreciates the adherence to STROBE and the Guidelines for reporting of statistics. The reviewer, with experience in tumor ablative techniques including thermal ablation, cryosurgery and irreversible electroporation, recognizes the value of these minimally invasive procedures. Therefore, the primary hypothesis that “ablative treatment is associated with lower risk of severe surgical complications” is well justified. This article, with detailed documentation of the length of hospital stay (LOS), rate of readmission and surgical complications, serves as a good reference for making informed decisions on tumor ablation vs. surgery on Small Renal Masses (T1a tumors).

The reviewer would like to raise a few questions and suggestions for the authors, to be specific:

1. The disease condition and treatment history of both groups can be valuable information to reference, as they can both affect the treatment outcome and the severity of compilations.

Disease condition including tumor stage (locally advanced or metastasized) or other complications before the treatment. A summary of treatment history including those received prior to the surgery/ablation procedure, adjuvant therapy and supportive/palliative therapy can be helpful too. For example, chemotherapy, radiation therapy and immunotherapy, even the choice of not going with active surveillance, can all negatively impact the quality of life of the patients.

2. Location of the tumors can be valuable information.

Typically, when deciding tumor ablation from surgery, the tumor location is an important factor to consider (at least at our center). For thermal ablation, the tumors are typically far from major blood vessels to avoid the “heat sink effect”. Therefore, the conclusion that “The most common surgical complications were bleeding and urinary leakage following surgery” can be largely attributed to the difference in tumor locations, not necessarily the invasive nature of nephrectomy.

3. The tumor recurrence comparison between surgery and ablation with the data can be a very helpful addition.

The authors stated that “In addition to better postoperative renal function, previous reports suggest that ablative therapies may be associated with similar rates of local recurrence-free survival, cancer-specific survival, metastasis-free survival, disease-free survival as well as a significantly reduced occurrence of post-operative complications compared to partial nephrectomy in T1a-tumors.” in the introduction. However, none of these analyses were mentioned or given in this study. The value of this article can be greatly elevated with these.

From the reviewer’s experience, since the ablative margin cannot be clearly identified during the RFA or MWA procedure, the risk of recurrence is always a major concern. The reviewer is eager to learn more from clinical experiences at other centers.

4. The tables, especially 2 to 5, may need to be better formatted. It looks to me that “tabs” are used to align the texts, which is a less-than-ideal approach (subject to formatting issue) than inserting a table with words staying within each cell. 

Round 2

Reviewer 1 Report

Comments and Suggestions for Authors

No further comments

Author Response

Once again - our gratitude for taking the time to re-assess the manuscript.

Reviewer 3 Report

Comments and Suggestions for Authors

Authors properly addressed reviewer's comments, overall improving quality of manuscript. P value of RENAL score in table 2 should be reported. 

Author Response

Once again - our gratitude for taking the time to re-assess the manuscript.

P-values have been added to the complexity scores in table 2.

Reviewer 4 Report

Comments and Suggestions for Authors

All comments have been addressed. 

Author Response

(The authors gave the same response as above.)
